# DARTS-: Robustly Stepping out of Performance Collapse Without Indicators

**Xiangxiang Chu**[1], **Xiaoxing Wang**[1,2]\*, **Bo Zhang**[1], **Shun Lu**[1,3]\*, **Xiaolin Wei**[1], **Junchi Yan**[2†]
[1]Meituan, [2]Shanghai Jiao Tong University, [3]University of Chinese Academy of Sciences
`{chuxiangxiang,zhangbo97,weixiaolin02}@meituan.com`
`{figure1_wxx,yanjunchi}@sjtu.edu.cn`
`lushun19@mails.ucas.ac.cn`

## ABSTRACT

Despite the fast development of differentiable architecture search (DARTS), it suffers from long-standing performance instability, which extremely limits its application. Existing robustifying methods draw clues from the resulting deteriorated behavior instead of finding out its causing factor. Various indicators such as Hessian eigenvalues are proposed as a signal to stop searching before the performance collapses. However, these indicator-based methods tend to easily reject good architectures if the thresholds are inappropriately set, let alone the searching is intrinsically noisy. In this paper, we undertake a more subtle and direct approach to resolve the collapse. We first demonstrate that skip connections have a clear advantage over other candidate operations, where it can easily recover from a disadvantageous state and become dominant. We conjecture that this privilege is causing degenerated performance. Therefore, we propose to factor out this benefit with an auxiliary skip connection, ensuring a fairer competition for all operations. We call this approach DARTS-. Extensive experiments on various datasets verify that it can substantially improve robustness. Our code is available at `https://github.com/Meituan-AutoML/DARTS-`.

## 1 INTRODUCTION

Recent studies (Zela et al., 2020; Liang et al., 2019; Chu et al., 2020b) have shown that one critical issue for differentiable architecture search (Liu et al., 2019b) regarding the performance collapse due to superfluous skip connections. Accordingly, some empirical indicators for detecting the occurrence of collapse have been produced. R-DARTS (Zela et al., 2020) shows that the loss landscape has more curvatures (characterized by higher Hessian eigenvalues w.r.t. architectural weights) when the derived architecture generalizes poorly. By regularizing for a lower Hessian eigenvalue, Zela et al. (2020); Chen & Hsieh (2020) attempt to stabilize the search process. Meanwhile, by directly constraining the number of skip connections to a fixed number (typically 2), the collapse issue becomes less pronounced (Chen et al., 2019b; Liang et al., 2019). These indicator-based approaches have several main drawbacks. Firstly, robustness relies heavily on the quality of the indicator. An imprecise indicator either inevitably accepts poor models or mistakenly reject good ones. Secondly, indicators impose strong priors by directly manipulating the inferred model, which is somewhat suspicious, akin to touching the test set. Thirdly, extra computing cost (Zela et al., 2020) or careful tuning of hyper-parameters (Chen et al., 2019b; Liang et al., 2019) are required. Therefore, it's natural to ask the following questions:

- Can we resolve the collapse without handcrafted indicators and restrictions to interfere with the searching and/or discretization procedure?

- Is it possible to achieve robustness in DARTS without tuning extra hyper-parameters?

---

\*Work done as an intern at Meituan Inc.
†Correspondent Author.

To answer the above questions, we propose an effective and efficient approach to stabilize DARTS. Our contributions can be summarized as follows:

**New Paradigm to Stabilize DARTS.** While empirically observing that current indicators (Zela et al., 2020; Chen & Hsieh, 2020) can avoid performance collapse at a cost of reduced exploration coverage in the search space, we propose a novel *indicator-free* approach to stabilize DARTS, referred to as DARTS-[1], which involves an auxiliary skip connection (see Figure 1) to remove the *unfair advantage* (Chu et al., 2020b) in the searching phase.

**Strong Robustness and Stabilization.** We conduct thorough experiments across seven search spaces and three datasets to demonstrate

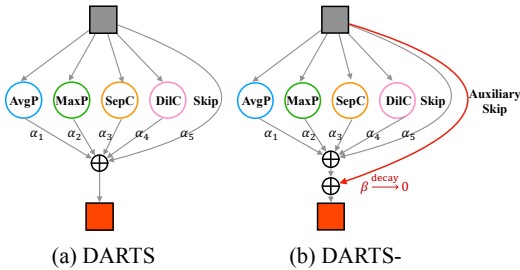

(a) DARTS    (b) DARTS-

Figure 1: Schematic illustration about (a) DARTS and (b) the proposed DARTS-, featuring an auxiliary skip connection (thick red line) with a decay rate $\beta$ between every two nodes to remove the potential unfair advantage that leads to performance collapse.

the effectiveness of our method. Specifically, our approach robustly obtains state-of-the-art results on 4 search space with $3\times$ fewer search cost than R-DARTS (Zela et al., 2020), which requires four independent runs to report the final performance.

**Seamless Plug-in Combination with DARTS Variants.** We conduct experiments to demonstrate that our approach can work together seamlessly with other orthogonal DARTS variants by removing their handcrafted indicators without extra overhead. In particular, our approach is able to improve 0.8% accuracy for P-DARTS, and 0.25% accuracy for PC-DARTS on CIFAR-10 dataset.

## 2 RELATED WORK

**Neural architecture search and DARTS variants.** Over the years, researchers have sought to automatically discover neural architectures for various deep learning tasks to relieve human from the tedious effort, ranging from image classification (Zoph et al., 2018), objection detection (Ghiasi et al., 2019), image segmentation (Liu et al., 2019a) to machine translation (So et al., 2019) etc. Among many proposed approaches, Differentiable Architecture Search (Liu et al., 2019b) features weight-sharing and resolves the searching problem via gradient descent, which is very efficient and easy to generalize. A short description of DARTS can be found in A.1. Since then, many subsequent works have been dedicated to accelerating the process (Dong & Yang, 2019b), reducing memory cost (Xu et al., 2020), or fostering its ability such as hardware-awareness (Cai et al., 2019; Wu et al., 2019), finer granularity (Mei et al., 2020) and so on. However, regardless of these endeavors, a fundamental issue of DARTS over its searching performance collapse remains not properly solved, which extremely prohibits its application.

**Robustifying DARTS.** As DARTS (Liu et al., 2019b) is known to be unstable as a result of performance collapse (Chu et al., 2020b), some recent works have devoted to resolving it by either designing indicators like Hessian eigenvalues for the collapse (Zela et al., 2020) or adding perturbations to regularize such an indicator (Chen & Hsieh, 2020). Both methods rely heavily on the indicator's accurateness, i.e., to what extent does the indicator correlate with the performance collapse? Other methods like Progressive DARTS (Chen et al., 2019b) and DARTS+ (Liang et al., 2019) employ a strong human prior, i.e., limiting the number of skip-connections to be a fixed value. Fair DARTS (Chu et al., 2020b) argues that the collapse results from the *unfair advantage* in an exclusive competitive environment, from which skip connections overly benefit to cause an abundant aggregation. To suppress such an advantage from overshooting, they convert the competition into collaboration where each operation is independent of others. It is however an indirect approach. SGAS (Li et al., 2020), instead, circumvents the problem with a greedy strategy where the unfair advantage can be prevented from taking effect. Nevertheless, potentially good operations might be pruned out too early because of greedy underestimation.

---

[1]We name it so as we undertake an inward way, as opposed to those outward ones who design new indicators, add extra cost and introduce new hyper-parameters.

## 3  DARTS-

### 3.1  MOTIVATION

We start from the detailed analysis of the role of skip connections. Skip connections were proposed to construct a residual block in ResNet  (He et al., 2016), which significantly improves training stability. It is even possible to deepen the network up to hundreds of layers without accuracy degradation by simply stacking them up. In contrast, stacking the plain blocks of VGG has degenerated performance when the network gets deeper. Besides, Ren et al. (2015); Wei et al. (2017); Tai et al. (2017); Li et al. (2018b) also empirically demonstrate that deep residual network can achieve better performance on various tasks.

From the gradient flow's perspective, skip connection is able to alleviate the gradient vanishing problem. Given a stack of $n$ residual blocks, the output of the $(i+1)^{\text{th}}$ residual block $X_{i+1}$ can be computed as $X_{i+1} = f_{i+1}(X_i, W_{i+1}) + X_i$, where $f_{i+1}$ denotes the operations of the $(i+1)^{\text{th}}$ residual block with weights $W_{i+1}$. Suppose the loss function of the model is $\mathcal{L}$, and the gradient of $X_i$ can be obtained as follows ($\mathbb{1}$ denotes a tensor whose items are all ones):

$$\frac{\partial \mathcal{L}}{\partial X_i} = \frac{\partial \mathcal{L}}{\partial X_{i+1}} \cdot \left( \frac{\partial f_{i+1}}{\partial X_i} + \mathbb{1} \right) = \frac{\partial \mathcal{L}}{\partial X_{i+j}} \cdot \prod_{k=1}^{j} \left( \frac{\partial f_{i+k}}{\partial X_{i+k-1}} + \mathbb{1} \right) \tag{1}$$

We observe that the gradient of shallow layers always includes the gradient of deep layers, which mitigates the gradient vanishing of $W_i$. Formally we have,

$$\frac{\partial \mathcal{L}}{\partial W_i} = \frac{\partial \mathcal{L}}{\partial X_{i+j}} \cdot \prod_{k=1}^{j} \left( \frac{\partial f_{i+k}}{\partial X_{i+k-1}} + \mathbb{1} \right) \cdot \frac{\partial f_i}{\partial W_i} \tag{2}$$

To analyze how skip connect affects the performance of residual networks, we introduce a trainable coefficient $\beta$ on all skip connections in ResNet. Therefore, the gradient of $X_i$ is converted to:

$$\frac{\partial \mathcal{L}}{\partial X_i} = \frac{\partial \mathcal{L}}{\partial X_{i+1}} \cdot \left( \frac{\partial f_{i+1}}{\partial X_i} + \beta \right) \tag{3}$$

Once $\beta < 1$, gradients of deep layers gradually vanish during the back-propagation (BP) towards shallow layers. Here $\beta$ controls the memory of gradients in BP to stabilize the training procedure.

We conduct a confirmatory experiment on ResNet50 and show the result in Fig 2. By initializing $\beta$ with $\{0, 0.5, 1.0\}$, we can visualize the tendency of $\beta$ along with training epochs. We observe that $\beta$ converges towards 1 after 40 epochs no matter the initialization, which demonstrates that the residual structure learns to push $\beta$ to a rather large value to alleviate gradient vanishing.

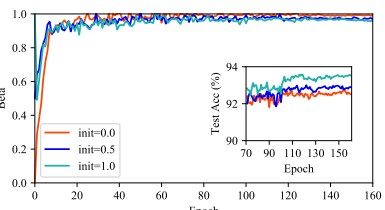

Similarly, DARTS (Liu et al., 2019b) utilizes a trainable parameter $\beta_{skip}$ to denote the importance of skip connection. However, In the search stage, $\beta_{skip}$ can generally increase and dominate the architecture parameters, and finally leads to performance collapse. We analyze that a large $\beta_{skip}$ in DARTS could result from two aspects: On the one hand, as the supernet automatically learns to alleviate gradient vanishing, it pushes $\beta_{skip}$ to a proper large value; On the other hand, the skip connection is indeed an important connection for the target network, which should be selected in the discretization stage. As

Figure 2: Tendency of trainable coefficient $\beta$ (initialized with $\{0, 0.5, 1\}$) of the skip connection in ResNet50 and test accuracy (inset figure) vs. epochs. The residual structure is proved to learn a large $\beta$ to ease training in all three cases. All models are trained and tested on CIFAR-10.

a consequence, the skip connection in DARTS plays two-fold roles: as *an auxiliary connection to stabilize the supernet training*, and as *a candidate operation to build the final network*. Inspired by the above observation and analysis, we propose to stabilize the search process by distinguishing the two roles of skip connection and handling the issue of gradient flow.

## 3.2 STEPPING OUT OF THE PERFORMANCE COLLAPSE

To distinguish the two roles, we introduce an auxiliary skip connection between every two nodes in a cell, see Fig. 1 (b). On the one hand, the fixed auxiliary skip connection carries the function of stabilizing the supernet training, even when $\beta_{skip}$ is rather small. On the other hand, it also breaks the unfair advantage (Chu et al., 2020b) as the advantageous contribution from the residual block is factored out. Consequently, the learned architectural parameter $\beta_{skip}$ can be freed from the role of controlling the memory of gradients, and is more precisely aimed to represent the relative importance of skip connection as a candidate operation. In contrast to Eq. 7, the output feature map of edge $e^{(i,j)}$ can now be obtained by Eq. 4, where $\beta_o^{i,j} = \frac{\exp(\alpha_o^{(i,j)})}{\sum_{o' \in \mathcal{O}} \exp(\alpha_{o'}^{(i,j)})}$ denotes the normalized importance, and $\beta$ is a coefficient independent from the architecture parameters.

Moreover, to eliminate the impact of auxiliary connection on the discretization procedure, we propose to decrease $\beta$ to 0 in the search phase, and our method can be degenerated to standard DARTS at the end of the search. Note that our method is insensitive to the type of decay strategy, so we choose linear decay by default for simplicity.

$$\bar{o}^{(i,j)}(x) = \beta x + \sum_{o \in \mathcal{O}} \beta_o^{(i,j)} o(x) = \left( \beta + \beta_{skip}^{(i,j)} \right) x + \sum_{o \neq skip} \beta_o^{(i,j)} o(x) \tag{4}$$

We then analyze how the auxiliary skip connection handles the issue of gradient flow. Referring to the theorem of a recent work by Zhou et al. (2020), the convergence of network weight $\mathbf{W}$ in the supernet can heavily depend on $\beta_{skip}$. Specifically, suppose only three operations (none, skip connection, and convolution) are included in the search space and MSE loss is utilized as training loss, when architecture parameters $\beta_{i,j}^o$ are fixed to optimize $\mathbf{W}$ via gradient descent, training loss can decrease by ratio $(1 - \eta\lambda/4)$ at one step with probability at least $1 - \delta$, where $\eta$ is the learning rate that should be bounded by $\delta$, and $\lambda$ follows Eq. 5.

$$\lambda \propto \sum_{i=0}^{h-2} \left[ \left( \beta_{conv}^{(i,h-1)} \right)^2 \prod_{t=0}^{i-1} \left( \beta_{skip}^{(t,i)} \right)^2 \right] \tag{5}$$

where $h$ is the number of layers of the supernet. From Eq. 5, we observe that $\lambda$ relies much on $\beta_{skip}$ than $\beta_{conv}$, which indicates that the network weights $\mathbf{W}$ can converge faster with a large $\beta_{skip}$. However, by involving an auxiliary skip connection weighted as $\beta$, Eq. 5 can be refined as follows:

$$\lambda \propto \sum_{i=0}^{h-2} \left[ \left( \beta_{conv}^{(i,h-1)} \right)^2 \prod_{t=0}^{i-1} \left( \beta_{skip}^{(t,i)} + \beta \right)^2 \right] \tag{6}$$

where $\beta \gg \beta_{skip}$ making $\lambda$ insensitive to $\beta_{skip}$, so that the convergence of network weights $\mathbf{W}$ depends more on $\beta_{conv}$. In the beginning of the search, the common value for $\beta_{skip}$ is 0.15 while $\beta$ is 1.0. From the view of convergence theorem (Zhou et al., 2020), the auxiliary skip connection alleviates the privilege of $\beta_{skip}$ and equalize the competition among architecture parameters. Even when $\beta$ gradually decays, the fair competition still holds since network weights $\mathbf{W}$ have been converged to an optimal point. Consequently, DARTS- is able to stabilize the search stage of DARTS.

Extensive experiments are performed to demonstrate the efficiency of the proposed auxiliary skip connection, and we emphasize that our method is flexible to combine with other methods to further improve the stabilization and searching performance. The overall algorithm is given in Alg. 1.

---

**Algorithm 1** DARTS-

**Require:**
    Network weights $w$; Architecture parameters $\alpha$;
    Number of search epochs $E$;
    Decay strategy for $\bar{\beta}_e, e \in \{1, 2, ..., E\}$.

**Ensure:**
    Searched architecture parameters $\alpha$.

1: Construct a super-network by stacking cells in which there is an auxiliary skip connection between every two nodes of choice
2: **for** each $e \in [1, E]$ **do**
3:     Update weights $w$ by $\nabla_w \mathcal{L}_{train}(w, \alpha, \beta_e)$
4:     Update parameters $\alpha$ by $\nabla_\alpha \mathcal{L}_{val}(w, \alpha, \beta_e)$
5: **end for**
6: Derive the final architecture based on learned $\alpha$ from the best validation supernet.

---

### 3.3 Relationship to Prior Work

Our method is aimed to address the performance collapse in differentiable neural architecture search. Most previous works (Zela et al., 2020; Chen & Hsieh, 2020; Liang et al., 2019) concentrate on developing various criteria or indicators characterizing the occurrence of collapse. Whereas, we don't study or rely on these indicators because they can mistakenly reject good models. Inspired by Chu et al. (2020b), our method focuses on calibrating the biased searching process. The underlying philosophy is simple: if the biased process is rectified, the searching result will be better. In summary, our method differs from others in two aspects: being process-oriented and indicator-free. Distinct from Chu et al. (2020b) that tweaks the competitive environment, our method can be viewed as one that breaks the unfair advantage. Moreover, we don't introduce any handcrafted indicators to represent performance collapse, thus greatly reducing the burden of shifting to different tasks.

## 4 Experiments

### 4.1 Search Spaces and Training Settings

For searching and evaluation in the standard DARTS space (we name it as **S0** for simplicity), we keep the same settings as in DARTS (Liu et al., 2019b). We follow R-DARTS (Zela et al., 2020) for their proposed reduced spaces **S1- S4** (harder than S0). However, the inferred models are trained with two different settings from R-DARTS (Zela et al., 2020) and SDARTS (Chen & Hsieh, 2020). The difference lies in the number of layers and initial channels for evaluation on CIFAR-100. R-DARTS sets 8 layers and 16 initial channels. Instead, SDARTS uses 20 and 36 respectively. For the proxyless searching on ImageNet, we instead search in MobileNetV2-like search space (we name it **S5**) proposed in FBNet (Wu et al., 2019). We use the SGD optimizer for weight and Adam ($\beta_1 = 0.5$ and $\beta_2 = 0.999$, 0.001 learning rate) for architecture parameters with the batch-size of 768. The initial learning rate is 0.045 and decayed to 0 within 30 epochs following the cosine schedule. We also use L2 regularization with 1e-4. It takes about 4.5 GPU days on Tesla V100. More details are provided in the appendix. We also use NAS-Bench-201 (**S6**) since DARTS performs severely bad. In total, we use 7 different search spaces to conduct the experiments, which involves three datasets.

### 4.2 Searching Results

**CIFAR-10 and CIFAR-100.** Following the settings of R-DARTS (Zela et al., 2020), we obtain an average top-1 accuracy of $97.41\%$ on CIFAR-10, as shown in Table 2. Moreover, our method is very robust since out of 5 independent runs the searching results are quite stable. The best cells found on

Table 1: Comparison of searched CNN in the DARTS search space on two different datasets.

| Dataset | DARTS | R-DARTS (L2) | Ours |
|---|---|---|---|
| C10 (S0) | 2.91±0.25 | 2.95±0.21 | **2.63±0.07** |
| C100 (S0) | 20.58±0.44 | 18.01±0.26 | **17.51±0.25** |

CIFAR-10 ($97.5\%$) are shown in Figure 9 (B). Results on CIFAR-100 are presented in Table 10 (see A.2.1). Moreover, our method has a much lower searching cost (**3× less**) than R-DARTS (Zela et al., 2020), where four independent searches with different regularization settings are needed to generate the best architecture. In other words, its robustness comes from the cost of more $CO_2$ emissions.

**ImageNet.** To further verify the efficiency of DARTS-, we directly search on ImageNet in S5 and compare our results with the state-of-the-art models under the mobile setting in Table 2. The visualization of the architecture is given in Fig 10. DARTS-A obtains $76.2\%$ top-1 accuracy on the ImageNet validation dataset. By contrast, direct applying DARTS on this search space only obtains $66.4\%$ (Chu et al., 2020b). Moreover, it obtains $77.8\%$ top-1 accuracy after being equipped with auto-augmentation (Cubuk et al., 2019) and squeeze-and-excitation (Hu et al., 2018), which are also used in EfficientNet.

**NAS-Bench-201.** Apart from standard search spaces, benchmarking with known optimal in a limited setting is also recommended. NAS-Bench-201 (Dong & Yang, 2020) consists of 15,625 architectures in a reduced DARTS-like search space, where it has 4 internal nodes and 5 operations per node. We compare our method with prior work in Table 3. We search on CIFAR-10 and look up the ground-truth performance with found genotypes on various test sets. Remarkably, we achieve a new state of the art, the best of which almost touches the optimal.

Table 2: Comparison of the state-of-the-art models on CIFAR-10 (left) and ImageNet (right). On CIFAR-10 dataset, our average result is obtained on 5 independently searched models to assure the robustness. For ImageNet, networks in the top block are directly searched on ImageNet; the middle indicates that architectures are searched on CIFAR-10 and then transferred to ImageNet; the bottom indicates models have SE and Swish. We search in S0 for CIFAR-10 and S5 for ImageNet.

| Models | Params (M) | FLOPs (M) | Acc (%) | Cost GPU Days |
|---|---|---|---|---|
| NASNet-A (2018) | 3.3 | 608[†] | 97.35 | 2000 |
| ENAS (2018) | 4.6 | 626[†] | 97.11 | 0.5 |
| DARTS (2019b) | 3.3 | 528[†] | 97.00±0.14[*] | 0.4 |
| SNAS (2019) | 2.8 | 422[†] | 97.15±0.02[*] | 1.5 |
| GDAS (2019b) | 3.4 | 519[†] | 97.07 | 0.2 |
| P-DARTS (2019b) | 3.4 | 532[†] | 97.5 | 0.3 |
| PC-DARTS (2020) | 3.6 | 558[†] | 97.43 | 0.1 |
| DARTS- (best) | 3.5 | 568 | 97.5 | 0.4 |
| P-DARTS (2019b)[‡] | 3.3±0.21 | 540±34 | 97.19±0.14 | 0.3 |
| R-DARTS (2020) | - | - | 97.05±0.21 | 1.6 |
| SDARTS-ADV (2020) | 3.3 | - | 97.39±0.02 | 1.3 |
| DARTS- (avg.) | 3.5±0.13 | 583±22 | 97.41±0.08 | 0.4 |

[†] Based on the provided genotypes.
[‡] 5 independent searches using their released code.
[*] Training the best searched model for several times (*whose average doesn't indicate the stability of the method*)

| Models | FLOPs (M) | Params (M) | Top-1 (%) | Top-5 (%) | Cost (GPU days) |
|---|---|---|---|---|---|
| AmoebaNet-A (2019) | 555 | 5.1 | 74.5 | 92.0 | 3150 |
| MnasNet-92 (2019) | 388 | 3.9 | 74.79 | 92.1 | 3791[‡] |
| FBNet-C (2019) | 375 | 5.5 | 74.9 | 92.3 | 9 |
| FairNAS-A (2019b) | 388 | 4.6 | 75.3 | 92.4 | 12 |
| SCARLET-C (2019a) | 365 | 6.7 | 76.9 | 93.4 | 10 |
| FairDARTS-D (2020b) | 440 | 4.3 | 75.6 | 92.6 | 3 |
| PC-DARTS (2020) | 597 | 5.3 | 75.8 | 92.7 | 3.8 |
| **DARTS- (ours)** | 467 | 4.9 | 76.2 | 93.0 | 4.5 |
| NASNet-A (2018) | 564 | 5.3 | 74.0 | 91.6 | 2000 |
| DARTS (2019b) | 574 | 4.7 | 73.3 | 91.3 | 0.4 |
| SNAS (2019) | 522 | 4.3 | 72.7 | 90.8 | 1.5 |
| PC-DARTS (2020) | 586 | 5.3 | 74.9 | 92.2 | 0.1 |
| FairDARTS-B (2020b) | 541 | 4.8 | 75.1 | 92.5 | 0.4 |
| MobileNetV3 (2019) | 219 | 5.4 | 75.2 | 92.2 | ≈3000 |
| MoGA-A (2020a) | 304 | 5.1 | 75.9 | 92.8 | 12 |
| MixNet-M (2019) | 360 | 5.0 | 77.0 | 93.3 | ≈3000 |
| EfficientNet B0 (2019) | 390 | 5.3 | 76.3 | 93.2 | ≈3000 |
| NoisyDARTS-A[◇] | 449 | 5.5 | 77.9 | 94.0 | 12 |
| **DARTS- (ours)**[◇] | 470 | 5.5 | 77.8 | 93.9 | 4.5 |

[‡] Estimated by Wu et al. (2019).
[◇] SE modules and Swish enabled.

Table 3: Searching performance on NAS-Bench-201 (Dong & Yang, 2020). Our method robustly obtains new SOTA. Averaged on 4 runs of searching. [1st]: first-order, [2nd]: second-order

| Method | Cost (hours) | CIFAR-10 | | CIFAR-100 | | ImageNet16-120 | |
|---|---|---|---|---|---|---|---|
| | | valid | test | valid | test | valid | test |
| DARTS[1st] (2019b) | 3.2 | 39.77±0.00 | 54.30±0.00 | 15.03±0.00 | 15.61±0.00 | 16.43±0.00 | 16.32±0.00 |
| DARTS[2nd] (2019b) | 10.2 | 39.77±0.00 | 54.30±0.00 | 15.03±0.00 | 15.61±0.00 | 16.43±0.00 | 16.32±0.00 |
| GDAS (2019b) | 8.7 | 89.89±0.08 | 93.61±0.09 | 71.34±0.04 | 70.70±0.30 | 41.59±1.33 | 41.71±0.98 |
| SETN (2019a) | 9.5 | 84.04±0.28 | 87.64±0.00 | 58.86±0.06 | 59.05±0.24 | 33.06±0.02 | 32.52±0.21 |
| DARTS- | 3.2 | **91.03±0.44** | **93.80±0.40** | **71.36±1.51** | **71.53±1.51** | **44.87±1.46** | **45.12±0.82** |
| DARTS- (best) | 3.2 | 91.55 | 94.36 | 73.49 | 73.51 | 46.37 | 46.34 |
| optimal | n/a | 91.61 | 94.37 | 73.49 | 73.51 | 46.77 | 47.31 |

**Transfer results on objection detection** We further evaluate the transfer-ability of our models on down-stream object detection task by replacing the backbone of RetinaNet (Lin et al., 2017) on MMDetection toolbox platform (Chen et al., 2019a). Specifically, with the same training setting as Chu et al. (2020b), our model achieves 32.5% mAP on the COCO dataset, surpassing other similar-sized models such as MobileNetV3, MixNet, and FairDARTS. The detailed results are shown in Appendix (Table 11).

## 4.3 ORTHOGONAL COMBINATION WITH OTHER VARIANTS

Our method can be flexibly adapted to combine with prior work for further improvements. Here we investigate the joint outcome with two methods: P-DARTS and PC-DARTS.

**Progressive DARTS (P-DARTS).** P-DARTS (Chen et al., 2019b) proposes a progressive approach to search gradually with deeper depths while pruning out the uncompetitive paths. Additionally, it makes use of some handcrafted criteria to address the collapse (the progressive idea itself cannot deal with it), for instance, they impose two strong priors by regularizing the number of skip connections $M$ as 2 as well as dropout. To be fair, we remove such a carefully handcrafted trick and run P-DARTS for several

Table 4: We remove the strong constraints on *the number of skip connections as 2 and dropout* (priors) for P-DARTS and compare its performance w/ and w/o DARTS-.

| Method | Setting | Acc (%) |
|---|---|---|
| P-DARTS | w/o priors | 96.48±0.55 |
| P-DARTS- | w/o priors | 97.28±0.04 |

Table 7: Comparison in various search spaces. We report the **lowest error rate** of 3 found architectures. [†]: under Chen & Hsieh (2020)'s training settings where all models have 20 layers and 36 initial channels (the best is shown in boldface). [‡]: under Zela et al. (2020)'s settings where CIFAR-100 models have 8 layers and 16 initial channels (The best is in boldface and underlined).

| Benchmark | | DARTS[‡] | R-DARTS[‡] | | DARTS[‡] | | Ours[‡] | PC-DARTS[†] | SDARTS[†] | | Ours[†] |
|---|---|---|---|---|---|---|---|---|---|---|---|
| | | | DP | L2 | ES | ADA | | | RS | ADV | |
| C10 | S1 | 3.84 | 3.11 | 2.78 | 3.01 | 3.10 | **2.68** | 3.11 | 2.78 | 2.73 | **2.68** |
| | S2 | 4.85 | 3.48 | 3.31 | 3.26 | 3.35 | **2.63** | 3.02 | 2.75 | 2.65 | **2.63** |
| | S3 | 3.34 | 2.93 | 2.51 | 2.74 | 2.59 | **2.42** | 2.51 | 2.53 | 2.49 | **2.42** |
| | S4 | 7.20 | 3.58 | 3.56 | 3.71 | 4.84 | **2.86** | 3.02 | 2.93 | 2.87 | **2.86** |
| C100 | S1 | 29.46 | 25.93 | 24.25 | 28.37 | 24.03 | **22.41** | 18.87 | 17.02 | **16.88** | 16.92 |
| | S2 | 26.05 | 22.30 | 22.24 | 23.25 | 23.52 | **21.61** | 18.23 | 17.56 | 17.24 | **16.14** |
| | S3 | 28.90 | 22.36 | 23.99 | 23.73 | 23.37 | **21.13** | 18.05 | 17.73 | 17.12 | **15.86** |
| | S4 | 22.85 | 22.18 | 21.94 | **21.26** | 23.20 | 21.55 | 17.16 | 17.17 | **15.46** | 17.48 |

times. As a natural control group, we also combine DARTS- with P-DARTS. We run both experiments for 3 times on CIFAR-10 dataset in Table 4. Without the strong priors, P-DARTS severely suffers from the collapse, where the inferred models contain an excessive number of skip connections. Specifically, it has a very high test error (3.42% on average), even worse than DARTS. However, P-DARTS can benefit greatly from the combination with DARTS-. The improved version (we call P-DARTS-) obtains much higher top-1 accuracy (**+0.8%**) on CIFAR-10 than its baseline.

**Memory Friendly DARTS (PC-DARTS).** To alleviate the large memory overhead from the whole supernet, PC-DARTS (Xu et al., 2020) selects the partial channel for searching. The proportion hyperparameter $K$ needs careful calibration to achieve a good result for specific tasks. As a byproduct, the search time is also reduced to 0.1 GPU days ($K$=4). We use their released code and run repeated experiments across different seeds under the same settings.

Table 5: Comparison of PC-DARTS *removing the strong prior* (i.e. channel shuffle) and combining DARTS-. The results are from 3 independent runs on CIFAR-10. The GPU memory cost is on a batch size of 256.

| Method | Setting | Acc (%) | Memory | Cost |
|---|---|---|---|---|
| PC-DARTS | $K = 2$ | 97.09±0.14 | 19.9G | 3.75h |
| PC-DARTS- | $K = 2$ | 97.35±0.02 | 20.8G | 3.41h |

To accurately evaluate the role of our method, we choose $K$=2 (a bad configuration in the original paper). We make comparisons between the original PC-DARTS and its combination with ours (named PC-DARTS-) in Table 5. PC-DARTS- can marginally boost the CIFAR-10 top-1 accuracy (+0.26% on average). The result also confirms that our method can make PC-DARTS less sensitive to its hyper-parameter $K$ while keeping its advantage of less memory cost and run time.

## 4.4 Ablation Study

**Robustness to Decay Strategy.** Our method is insensitive to the type of decay policy on $\beta$. We design extra two strategies as comparisons: *cosine* and *step* decay. They both have the similar performance. Specifically, when $\beta$ is scheduled to zero by the cosine strategy, the average accuracy of four searched CIFAR-10 models in S3 is 97.33%±0.09, the best is 97.47%. The step decay at epoch 45 obtains 97.30% top-1 accuracy on average in the same search space.

**Robustness Comparison on C10 and C100 in S0-S4.** To verify the robustness, it is required to search several times to report the average performance of derived models (Yu et al., 2020; Antoine et al., 2020). As shown in Table 1, Table 9 and Table 7, our method outperforms the recent SOTAs across several spaces and datasets. Note that SDARTS-ADV utilizes adversarial training and requires $3\times$ more search time than ours. Especially, we find a good model in $S_3$ on CIFAR-100 with the lowest top-1 test error 15.86%. The architectures of these models can be found in the appendix.

**Sensitivity Analysis of** $\beta$ The power of the auxiliary skip connection branch can be discounted by setting a lower initial $\beta$. We now evaluate how sensitive

Table 6: Searching performance on CIFAR-10 in S3 w.r.t the initial linear decay rate $\beta_0$. Each setting is run for three times.

| $\beta_0$ | Error (%) |
|---|---|
| 1 | 2.65±0.04 |
| 0.7 | 2.76±0.16 |
| 0.4 | 3.04±0.19 |
| 0.1 | 3.11±0.16 |
| 0 | 4.58±1.3 |

our approach is to the value of $\beta$. It's trivial to see that our approach degrades to DARTS when $\beta = 0$. We compare results when searching with $\beta \in \{1, 0.7, 0.4, 0.1, 0\}$ in Table 6, which show that a bigger $\beta_0$ is advantageous to obtain better networks.

**The choice of auxiliary branch** Apart from the default skip connection serving as the auxiliary branch, we show that it is also effective to replace it with a learnable 1×1 convolution projection, which is initialized with an identity tensor. The average accuracy of three searched CIFAR-10 models in S3 is 97.25%±0.09. Akin to the ablation in He et al. (2016), the projection convolution here works in a similar way as the proposed skip connection. This proves the necessity of the auxiliary branch.

**Performance with longer epochs.** It is claimed by Bi et al. (2019) that much longer epochs lead to better convergence of the supernet, supposedly beneficial for inferring the final models. However, many of DARTS variants fail since their final models are full of skip connections. We thus evaluate how our method behaves in such a situation. Specifically, we extend the standard 50 epochs to 150, 200 and we search 3 independent times each for S0, S2 and S3. Due to the longer epochs, we slightly change our decay strategy, we keep $\beta = 1$ all the way until for last 50 epochs we decay $\beta$ to 0. Other hyper-parameters are kept unchanged. The results are shown in Table 8 and the found genotypes are listed in Figure 17, 18, 19, 20 and 21. It indicates that DARTS- doesn't suffer from longer epochs since it has reasonable values for $\#P$ compared with those ($\#P = 0$) investigated by Bi et al. (2019). Notice S2 and S3 are harder cases where DARTS suffers more severely from the collapse than S0. As a result, *DARTS- can successfully survive longer epochs even in challenging search spaces.* Noticeably, it is still unclear whether longer epochs can truly boost searching performance. Although we achieve a new state-of-the-art result in S2 where the best model has 2.50% error rate (previously 2.63%), it still has worse average performance (2.71±0.11%) in S0 than the models searched with 50 epochs (2.59±0.08%), and the best model in S3 (2.53%) is also weaker than before (2.42%).

Table 8: Searching performance on CIFAR-10 in S0, S2 and S3 using longer epochs. Following Bi et al. (2019), $\#P$ means the number of parametric operators in the normal cell. Averaged on 3 runs of search.

| Epoch | S0 | | | S2 | | | S3 | | |
|---|---|---|---|---|---|---|---|---|---|
| | $\#P$ | Params (M) | Error (%) | $\#P$ | Params (M) | Error (%) | $\#P$ | Params (M) | Error (%) |
| 150 | 6.6±1.1 | 3.3±0.3 | 2.74±0.06 | 6.0±0.0 | 3.9±0.3 | 2.58±0.11 | 6.0±1.0 | 3.6±0.3 | 2.55±0.03 |
| 200 | 7.3±0.6 | 3.2±0.3 | 2.71±0.11 | 8.0±0.0 | 4.3±0.1 | 2.65±0.21 | 7.6±0.5 | 4.3±0.2 | 2.66±0.09 |

Besides, compared with first-order DARTS with a cost of 0.4 GPU days in S0, Amended-DARTS (Bi et al., 2019), particularly designed to survive longer epochs, reports 1.7 GPU days even with pruned edges in S0. Our approach has the same cost as first-order DARTS, which is more efficient.

## 5 ANALYSIS AND DISCUSSIONS

### 5.1 FAILURE OF HESSIAN EIGENVALUE

The maximal Hessian eigenvalue calculated from the validation loss w.r.t $\alpha$ is regarded as an indicator of performance collapse (Zela et al., 2020; Chen & Hsieh, 2020). Surprisingly, our method develops a growing eigenvalue in the majority of configurations, which conflicts with the previous observations. We visualize these statistics across different search space and datasets in Figure 4 (A.2.2). Although eigenvalues increase almost monotonously and reach a relatively large value in the end, the final models still have good performance that matches with state-of-the-art ones (see Table 9). These models can be mistakenly deemed as bad ones or never visited according to the eigenvalue criteria. Our observations disclose one fatal drawback of these indicator-based approaches: *they are prone to rejecting good models.* Further analysis can be found in A.2.2.

## 5.2 VALIDATION ACCURACY LANDSCAPE

Recent works, R-DARTS (Zela et al., 2020) and SDARTS (Chen & Hsieh, 2020) point that the architectural weights are expected to converge to an optimal point where accuracy is insensitive to perturbations to obtain a stable architecture after the discretization process, i.e., the convergence point should have a smooth landscape. SDARTS proposes a perturbation-based regularization, which further stabilizes the searching process of DARTS. However, the perturbation regularization disturbs the training procedure and thus misleads the update of architectural weights. Different from SDARTS that explicitly smooths landscape by perturbation, DARTS- can implicitly do the same without directly perturbing architectural weights.

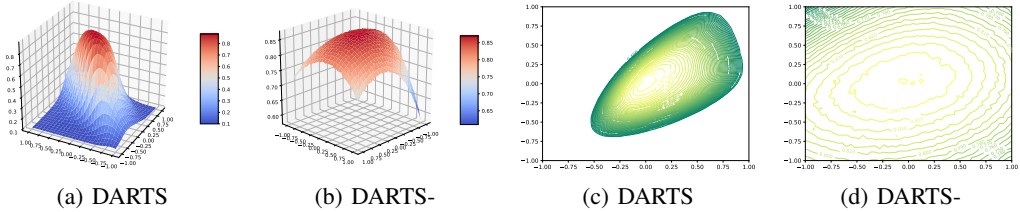

| (a) DARTS | (b) DARTS- | (c) DARTS | (d) DARTS- |

Figure 3: Comparison of the validation accuracy landscape of (a) DARTS and (b) DARTS- w.r.t. $\alpha$ on CIFAR-10 in S3. Their contour maps are shown respectively in (c) and (d), where we set the step of contour map as 0.1. The accuracy of the derived models are 94.84% (a,c) and 97.58% (b,d), while the maximum Hessian eigenvalues are similarly high (0.52 and 0.42)

To analyze the efficacy of DARTS-, we plot the validation accuracy landscape w.r.t architectural weights $\alpha$, and find that auxiliary connection smooths the landscape and thus stabilizes the searching stage. Specifically, we choose two random directions and apply normalized perturbation on $\alpha$ (following Li et al. 2018a). As shown in Figure 3, DARTS- is less sensitive to the perturbation than DARTS, and the contour map of DARTS- descends more gently.

## 6 CONCLUSION

We propose a simple and effective approach named DARTS- to address the performance collapse in differentiable architecture search. Its core idea is to make use of an auxiliary skip connection branch to take over the gradient advantage role from the candidate skip connection operation. This can create a fair competition where the bi-level optimization process can easily differentiate good operations from the bad. As a result, the search process is more stable and the collapse seldom happens across various search spaces and different datasets. Under strictly controlled settings, it steadily outperforms recent state-of-the-art RobustDARTS (Zela et al., 2020) with 3× fewer search cost. Moreover, our method disapproves of various handcrafted regularization tricks. Last but not least, it can be used stand-alone or in cooperation with various orthogonal improvements if necessary.

This paper conveys two important messages for future research. On the one hand, the Hessian eigenvalue indicator for performance collapse (Zela et al., 2020; Chen & Hsieh, 2020) is not ideal because it has a risk of rejecting good models. On the other hand, handcraft regularization tricks (Chen et al., 2019b) seem more critical to search a good model instead of the proposed methods. Then what's the solution? In principle, it's difficult to find a perfect indicator of the collapse. Our approach shows the potential to control the search process and doesn't impose limitations or priors on the final model. We hope more attention be paid in this direction.

## 7 ACKNOWLEDGEMENT

This research was supported by Meituan.

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

## A APPENDIX

### A.1 PRELIMINARY ABOUT DARTS

In differentiable architecture search (Liu et al., 2019b), a cell-based search space in the form of Directed Acyclic Graph (DAG) is constructed. The DAG has two input nodes from the previous layers, four intermediate nodes, and one output node. There are several paralleling operators (denoted as $\mathcal{O}$) between each two nodes (say $i$, $j$), whose output $\bar{o}^{(i,j)}$ given an input $x$ is defined as,

$$\bar{o}^{(i,j)}(x) = \sum_{o \in \mathcal{O}} \frac{\exp(\alpha_o^{(i,j)})}{\sum_{o' \in \mathcal{O}} \exp(\alpha_{o'}^{(i,j)})} o(x) \tag{7}$$

It is essentially applying softmax over all operators where each operator is assigned with an architectural weight $\alpha$. A supernet is built on two kinds of such cells, so-called normal cells and reduction cells (for down-sampling). The architectural search is then characterized as a bi-level optimization:

$$\min_{\alpha} \quad \mathcal{L}_{val}(w^*(\alpha), \alpha) \tag{8}$$

$$s.t. \quad w^*(\alpha) = \arg\min_{w} \mathcal{L}_{train}(w, \alpha) \tag{9}$$

This indicates that the training of such a cell-based supernet should be interleaved, where at each step the network weights and architectural weights are updated iteratively. The final model is determined by simply choosing operations with the largest architectural weights.

### A.2 EXPERIMENT

#### A.2.1 TRAINING DETAILS

**CIFAR-10 and CIFAR-100.** Table 9 gives the averaged performance in reduced search spaces S1-S4, as well with maximum eigenvalues. Table 10 reports the CIFAR-100 results in S0.

Table 9: Comparison of searched CNN architectures in four reduced search spaces S1-S4 (Zela et al., 2020) on CIFAR-10 and CIFAR-100. We report the mean±std of test error over 3 found architectures retrained from scratch, alongside with eigenvalue (EV) that corresponds to the best validation accuracy. We follow the same settings as Zela et al. (2020).

| Benchmark | | DARTS | DARTS-ES | DARTS-ADA | ours | ours (EV) |
|---|---|---|---|---|---|---|
| C10 | S1 | 4.66±0.71 | 3.05±0.07 | 3.03±0.08 | **2.76±0.07** | 0.37±0.19 |
| | S2 | 4.42±0.40 | 3.41±0.14 | 3.59±0.31 | **2.79±0.04** | 0.41±0.08 |
| | S3 | 4.12±0.85 | 3.71±1.14 | 2.99±0.34 | **2.65±0.04** | 0.31±0.09 |
| | S4 | 6.95±0.18 | 4.17±0.21 | 3.89±0.67 | **2.91±0.04** | 0.30±0.11 |
| C100 | S1 | 29.93±0.41 | 28.90±0.81 | 24.94±0.81 | **23.26±0.59** | 0.60±0.15 |
| | S2 | 28.75±0.92 | 24.68±1.43 | 26.88±1.11 | **22.31±0.65** | 0.54±0.15 |
| | S3 | 29.01±0.24 | 26.99±1.79 | 24.55±0.63 | **21.47±0.40** | 0.40±0.03 |
| | S4 | 24.77±1.51 | 23.90±2.01 | 23.66±0.90 | **21.75±0.26** | 0.93±0.15 |

**ImageNet classification.** For training on ImageNet, we use the same setting as MnasNet (Tan et al., 2019). To be comparable with EfficientNet (Tan & Le, 2019), we also use squeeze-and-excitation (Hu et al., 2018). Furthermore, we don't include methods trained using large model distillation, because it can boost final validation accuracy marginally. To be fair, we don't use the efficient head in MobileNetV3 (Howard et al., 2019) although it can reduce FLOPs marginally.

**COCO object detection.** All models are trained and evaluated on MS COCO dataset for 12 epochs with a batch size of 16. The initial learning rate is 0.01 and reduced by 0.1 at epoch 8 and 11.

#### A.2.2 FURTHER DISCUSSIONS ON FAILURE OF EIGENVALUE

To further explore the relationship between the searching performance and Hessian Eigenvalue, we plot the performance trajectory of the searched models in Figure 4 (b). Specifically, we sample

Table 10: Comparison of searched models on CIFAR-100. $\diamond$: Reported by Dong & Yang (2019b), $\star$: Reported by Zela et al. (2020), $\ddagger$:Rerun their code.

| Models | Params (M) | Error (%) | Cost GPU Days |
|---|---|---|---|
| ResNet (2016) | 1.7 | 22.10$^\diamond$ | - |
| AmoebaNet (2019) | 3.1 | 18.93$^\diamond$ | 3150 |
| PNAS (2018) | 3.2 | 19.53$^\diamond$ | 150 |
| ENAS (2018) | 4.6 | 19.43$^\diamond$ | 0.45 |
| DARTS (2019b) | - | 20.58±0.44$^\star$ | 0.4 |
| GDAS (2019b) | 3.4 | 18.38 | 0.2 |
| P-DARTS (2019b) | 3.6 | 17.49$^\ddagger$ | 0.3 |
| R-DARTS (2020) | - | 18.01±0.26 | 1.6 |
| DARTS- (avg.) | 3.3 | 17.51±0.25 | 0.4 |
| **DARTS- (best)** | 3.4 | 17.16 | 0.4 |

Table 11: Transfer results on COCO datasets of various drop-in backbones.

| Backbones | Params (M) | Acc | AP | $AP_{50}$ | $AP_{75}$ | $AP_S$ | $AP_M$ | $AP_L$ |
|---|---|---|---|---|---|---|---|---|
| MobileNetV2 (2018) | 3.4 | 72.0 | 28.3 | 46.7 | 29.3 | 14.8 | 30.7 | 38.1 |
| SingPath NAS (2019) | 4.3 | 75.0 | 30.7 | 49.8 | 32.2 | 15.4 | 33.9 | 41.6 |
| MnasNet-A2 (2019) | 4.8 | 75.6 | 30.5 | 50.2 | 32.0 | 16.6 | 34.1 | 41.1 |
| MobileNetV3 (2019) | 5.4 | 75.2 | 29.9 | 49.3 | 30.8 | 14.9 | 33.3 | 41.1 |
| MixNet-M (2019) | 5.0 | 77.0 | 31.3 | 51.7 | 32.4 | 17.0 | 35.0 | 41.9 |
| FairDARTS-C (2020b) | 5.3 | 77.2 | 31.9 | 51.9 | 33.0 | 17.4 | 35.3 | 43.0 |
| **DARTS-A (Ours)** | 5.5 | 77.8 | 32.5 | 52.8 | 34.1 | 18.0 | 36.1 | 43.4 |

models every 10 epochs and train these models from scratch using the same setting as above (Figure 5). The performance of the inferred models continues growing, where the accuracy is boosted from 96.5% to 97.4%. This affirms the validity of searching using our method. In contrast, the early-stopping strategies based on eigenvalues (Zela et al., 2020) would fail in this setting. We argue that the proposed auxiliary skip branch can regularize the overfitting of the supernet, leaving the architectural weights to represent the ability of candidate operations. This experiment poses as a counterexample to R-DARTS, where good models can appear albeit Hessian eigenvalues change fast. It again denies the need for a costly indicator.

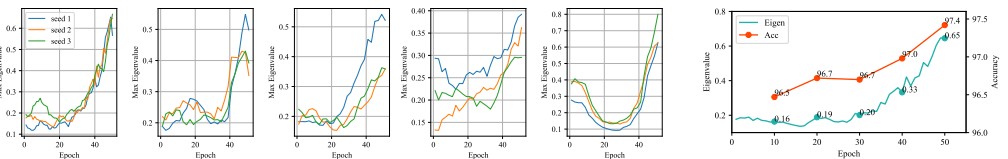

(a) Trajectory of eigenvalues in S0-S4 (left to right) on CIFAR-10   (b) Sampled models' performance

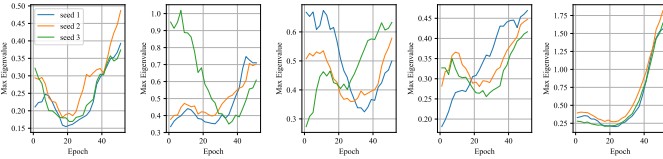

(c) Trajectory of eigenvalues in S0-S4 (left to right) on CIFAR-100

Figure 4: The evolution of maximal eigenvalues of DARTS- when searching in different search spaces S0-S4 on CIFAR-10 (a) and CIFAR-100 (c). We run each experiment 3 times on different seeds. (b) DARTS-'s growing Hessian eigenvalues don't induce poor performance. Among the sampled five models, the one corresponding to the highest eigenvalue has the best performance. This example is done in S0 on CIFAR-10.

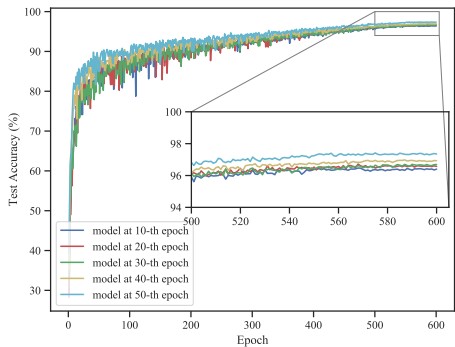

Figure 5: Training five models sampled every 10 epochs during DARTS- searching process. See Fig 4 (b) for the corresponding eigenvalues.

### A.2.3 MORE ABLATION STUDIES

To supplement the sensitivity analysis in Section 4.4, Figure 6 shows the training loss curve when initializing $\beta$ with different values.

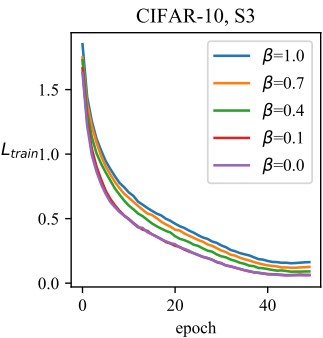

Figure 6: The training loss curve of the over-parameterized network on CIFAR-10 in S3 with different initial $\beta_0$.

### A.3 LOSS LANDSCAPE

To show the consistent smoothing capacity of DARTS-, we draw more loss landscapes in S0 and S3 (on several seeds) w.r.t architectural parameters and their contours in Figure 7 and 8. Generally, DARTS-'s slopes are more inflated if we consider them as camping tents, which suggest better convergence of the over-parameterized network.

### A.4 LIST OF EXPERIMENTS

We summarize all the conducted experiments with their related figures and tables in Table 12.

## B FIGURES OF GENOTYPES

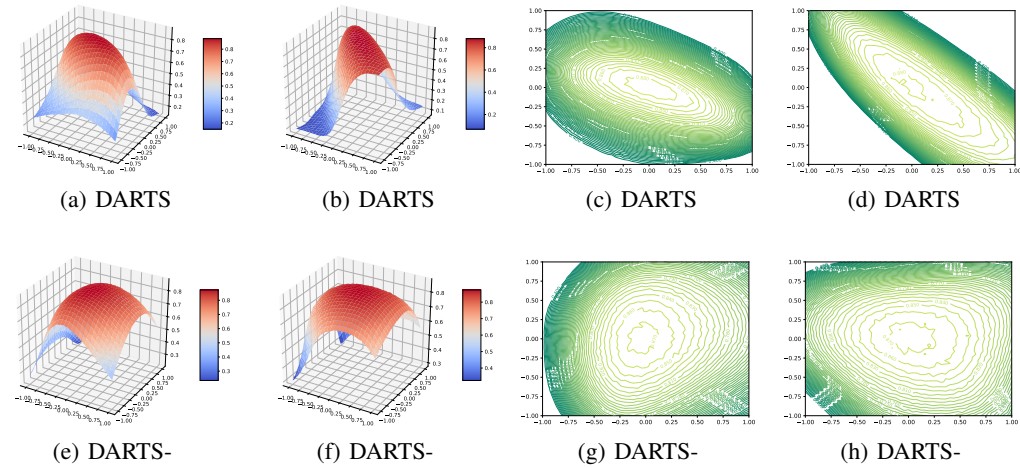

Figure 7: More visualization of validation accuracy landscapes of DARTS (a,b) and DARTS- (e,f) w.r.t. the architectural weights $\alpha$ on CIFAR-10 in S0. Their contour maps are shown respectively in (c,d) and (g,h). The step of contour map is 0.1. The inferred models by DARTS- have higher accuracies (97.50%, 97.49%) than DARTS (97.19%, 97.20%).

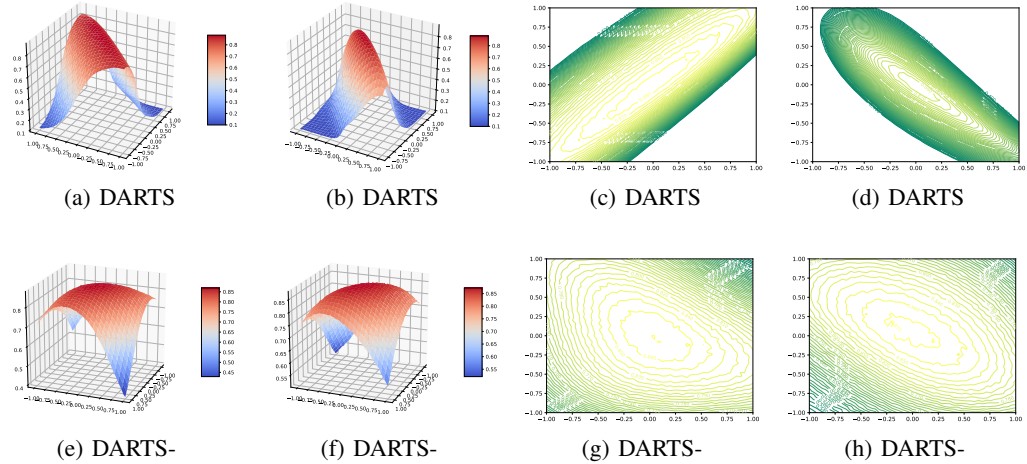

Figure 8: More visualization of validation accuracy landscapes of DARTS (a,b) and DARTS- (e,f) w.r.t. the architectural weights $\alpha$ on CIFAR-10 in S3. Their contour maps are shown respectively in (c,d) and (g,h). The step of contour map is 0.1.

Table 12: List of experiments conducted in this paper

| Method | Search Space | Dataset | Figures | | Tables | |
| --- | --- | --- | --- | --- | --- | --- |
| | | | Main text | Supp. | Main text | Supp. |
| DARTS | S0 | CIFAR-10 | | 7 | | |
| DARTS | S3 | CIFAR-10 | 3 | | | |
| DARTS- | S0 | CIFAR-10 | | 4,5,9,7 | 1,2 | |
| DARTS- | S1-S4 | CIFAR-10 | 3 | 4,6,9,8 | 7,9 | 6 |
| DARTS- | S5 | ImageNet | | 5,10 | 2 | |
| DARTS- | S5 | MS COCO | | | 11 | |
| DARTS- | S6 | CIFAR-10 | | | 3 | |
| DARTS- | S1-S4 | CIFAR-100 | | 4,11 | 7 | 9 |
| DARTS- | S0 | CIFAR-100 | | 4,11 | 1,10 | |
| P-DARTS w/o M=2 | S0 | CIFAR-10 | | 12 | 4 | |
| P-DARTS w/ auxiliary skip | S0 | CIFAR-10 | | 13 | 4 | |
| PC-DARTS w/o channel shuffling | S0 | CIFAR-10 | | 14 | 5 | |
| PC-DARTS w/ auxiliary skip | S0 | CIFAR-10 | | 15 | 5 | |

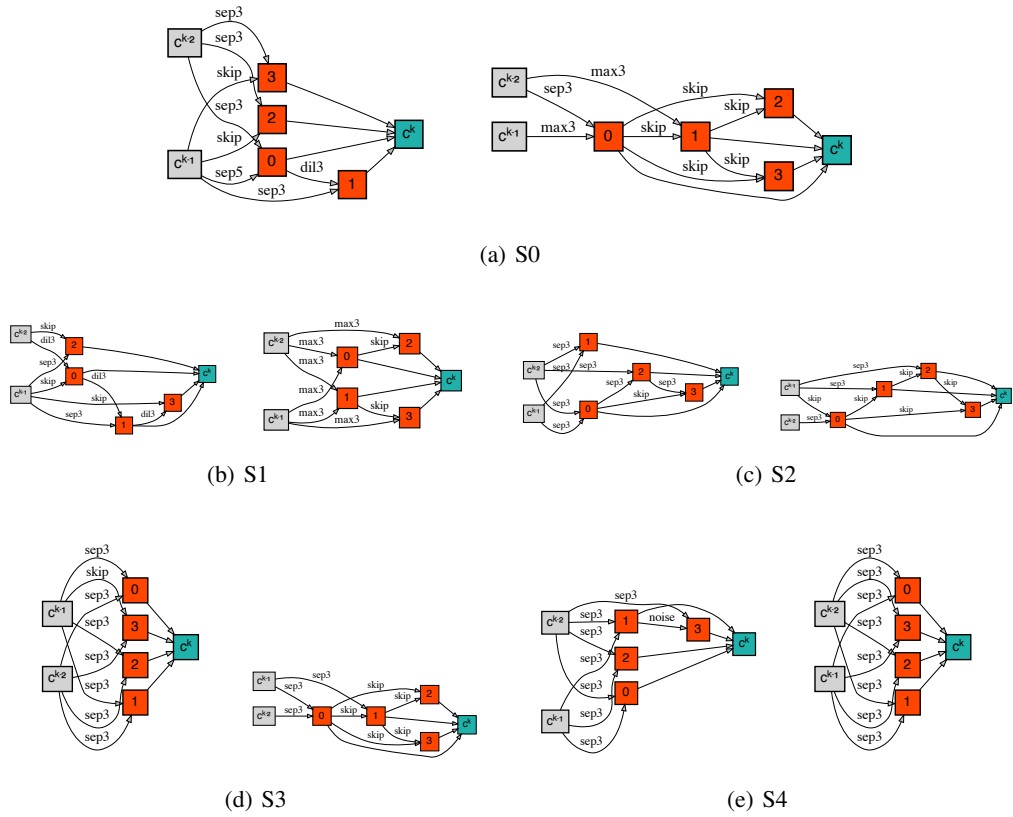

Figure 9: The best found normal cell and reduction cell in search spaces S0-S4 on CIFAR-10 dataset.

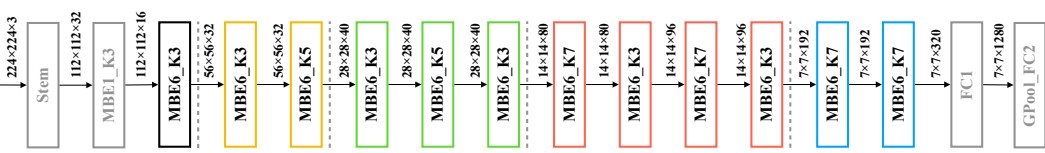

Figure 10: Architecture of DARTS-A searched on ImageNet dataset.

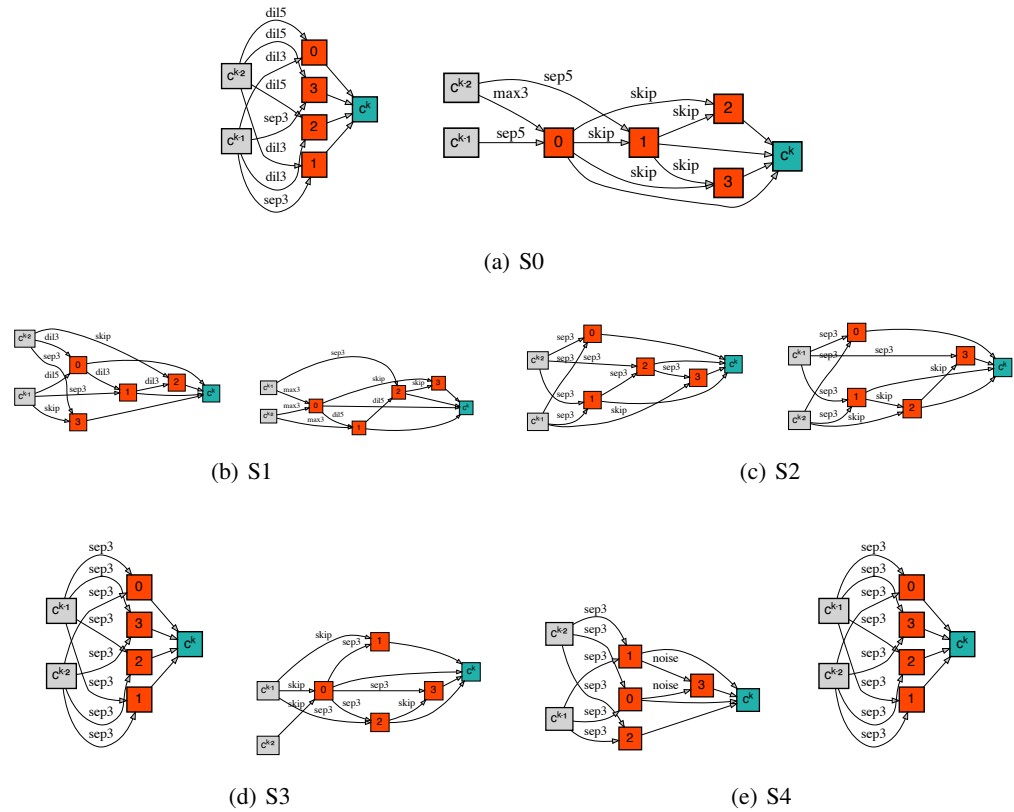

Figure 11: The best found normal cell and reduction cell in search spaces S0-S4 on CIFAR-100 dataset.

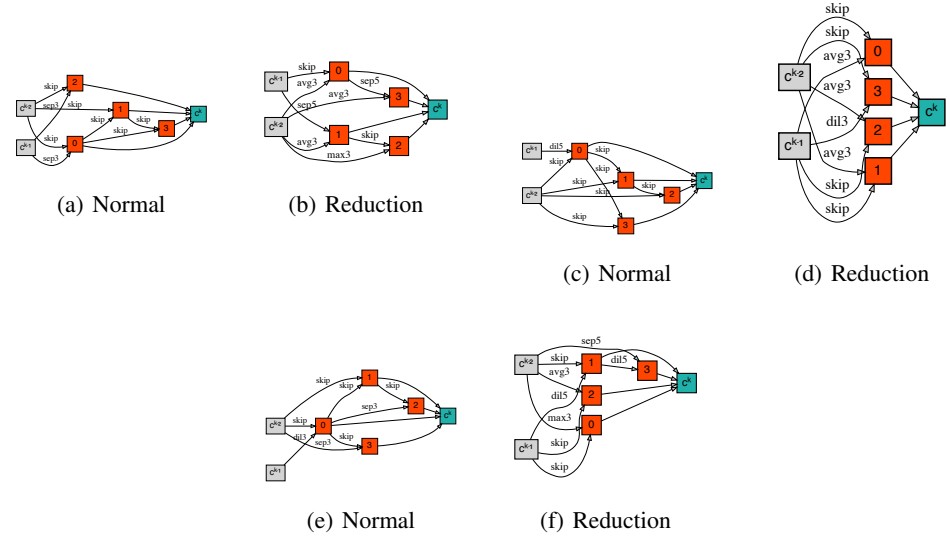

Figure 12: Found normal cells and reduction cells by P-DARTS (Chen et al., 2019b) without prior (M=2) in the DARTS' standard search space on CIFAR-10 dataset.

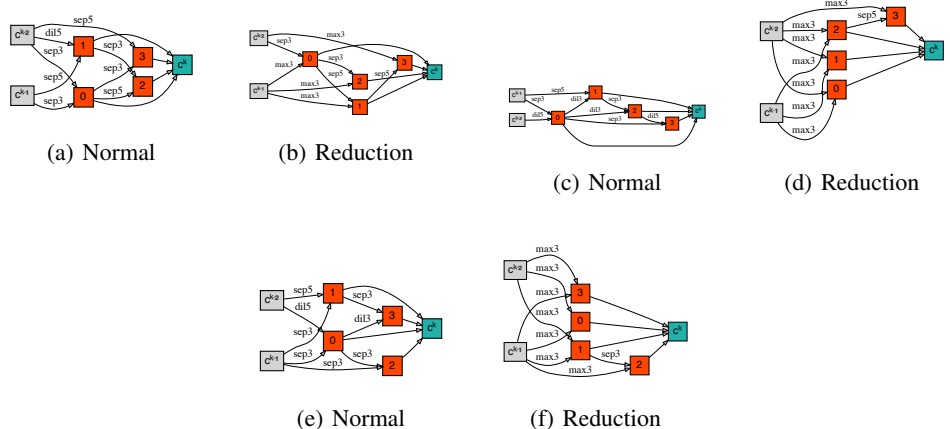

Figure 13: Found normal cells and reduction cells by P-DARTS (Chen et al., 2019b) with the proposed auxiliary skip connections in the DARTS' standard search space on CIFAR-10 dataset.

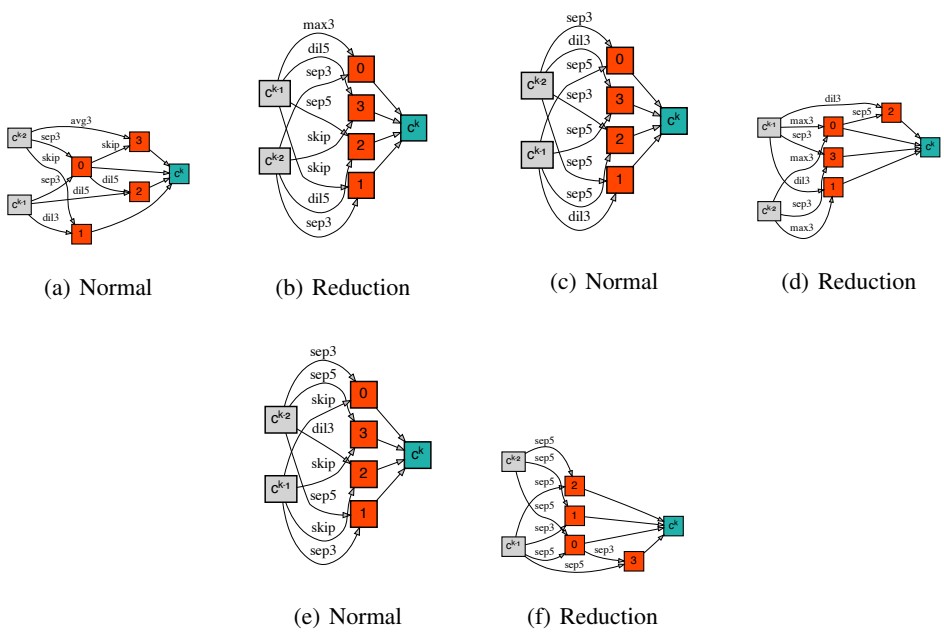

Figure 14: Found normal cells and reduction cells by PC-DARTS (Xu et al., 2020) without channel shuffling in the DARTS' standard search space on CIFAR-10 dataset.

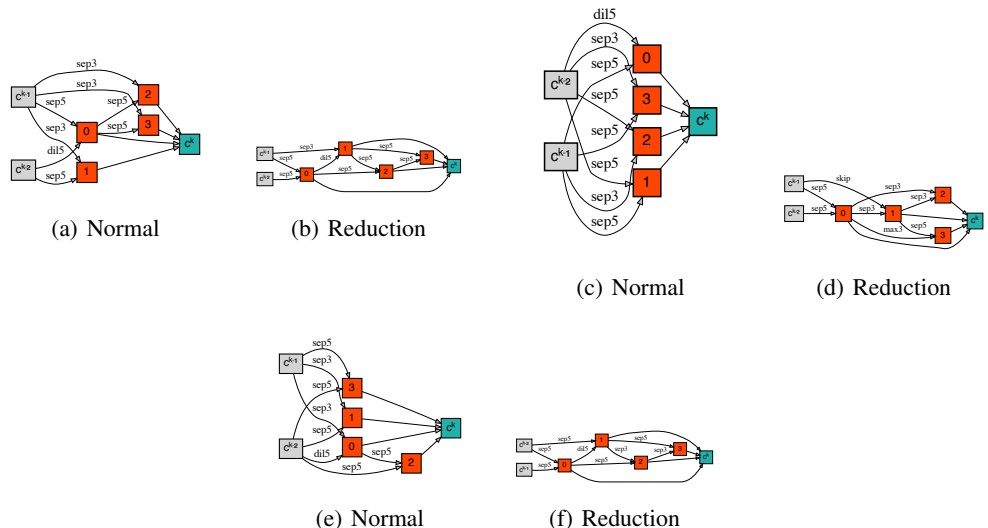

Figure 15: Found normal cells and reduction cells by PC-DARTS (Xu et al., 2020) with the proposed auxiliary skip connections in the DARTS' standard search space on CIFAR-10 dataset.

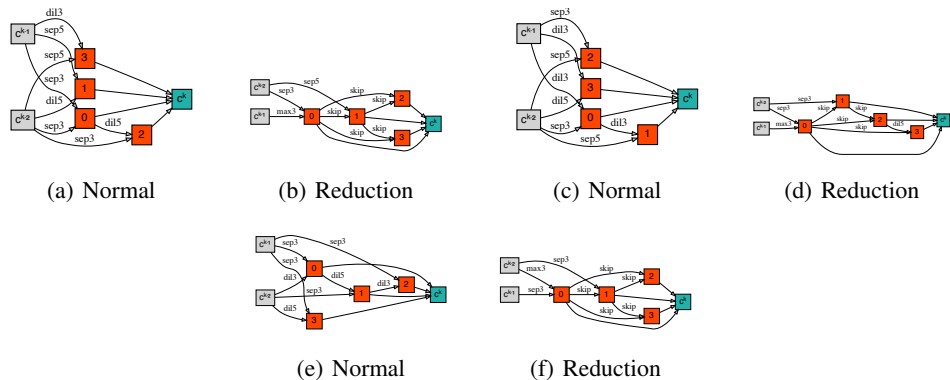

Figure 16: Keep $\beta_{skip} = 1$ throughout the DARTS- searching in the DARTS' standard search space on CIFAR-10 dataset.

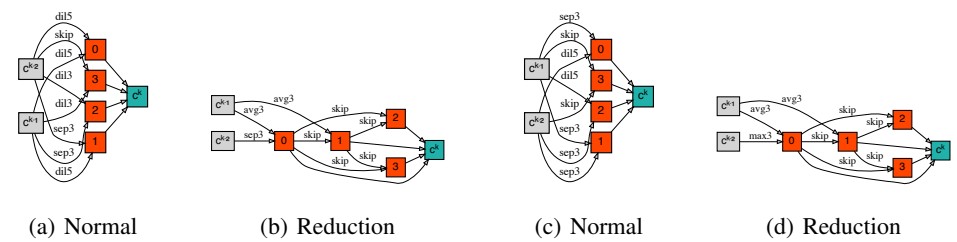

Figure 17: Best cells found when decaying $\beta_{skip}$ in the last 50 epochs during the DARTS- searching for 150 and 200 epochs respectively in the DARTS search space on CIFAR-10.

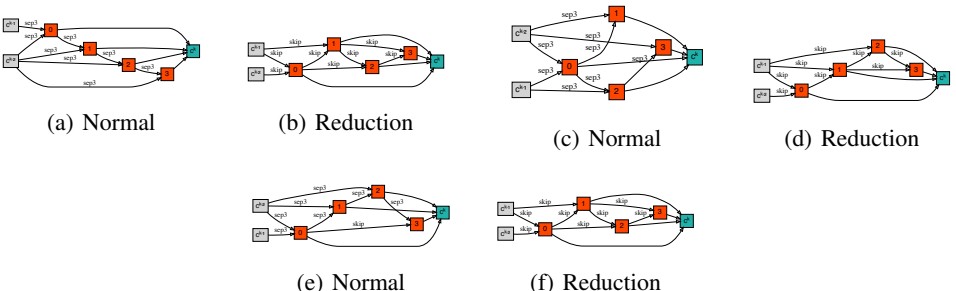

(a) Normal      (b) Reduction      (c) Normal      (d) Reduction

(e) Normal      (f) Reduction

Figure 18: Decaying $\beta_{skip}$ in the last 50 epochs during the DARTS- searching for 150 epochs in S2 on CIFAR-10 dataset.

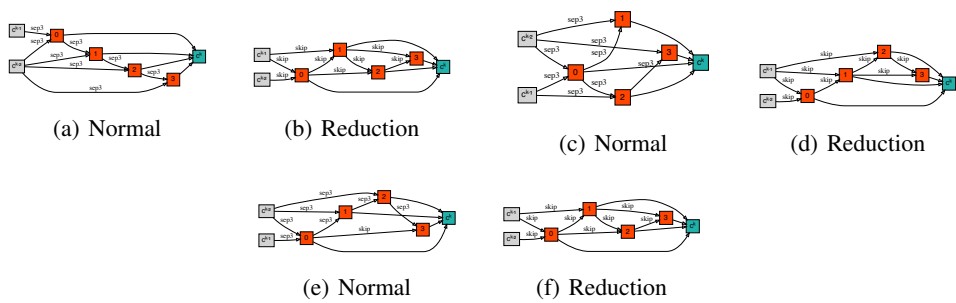

(a) Normal      (b) Reduction      (c) Normal      (d) Reduction

(e) Normal      (f) Reduction

Figure 19: Decaying $\beta_{skip}$ in the last 50 epochs during the DARTS- searching for 200 epochs in S2 on CIFAR-10 dataset.

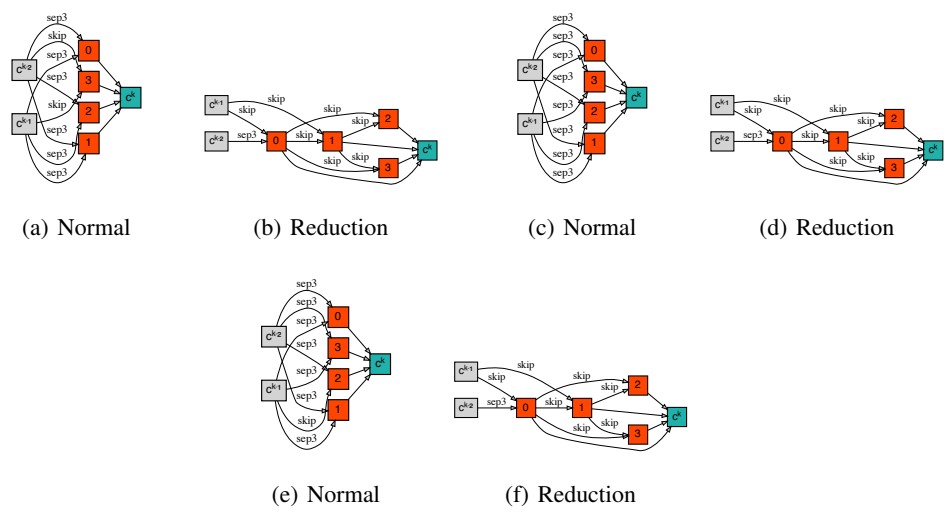

(a) Normal      (b) Reduction      (c) Normal      (d) Reduction

(e) Normal      (f) Reduction

Figure 20: Decaying $\beta_{skip}$ in the last 50 epochs during the DARTS- searching for 150 epochs in S3 on CIFAR-10 dataset.

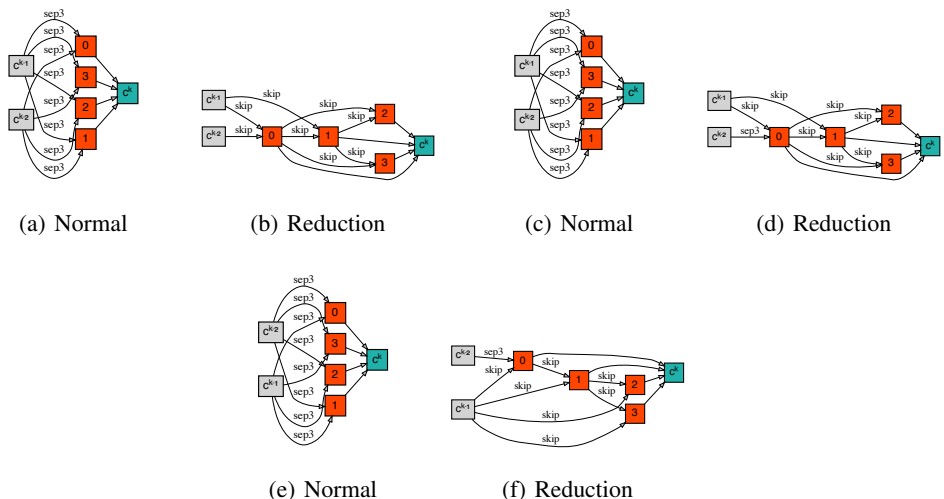

(a) Normal      (b) Reduction      (c) Normal      (d) Reduction

(e) Normal      (f) Reduction

Figure 21: Decaying $\beta_{skip}$ in the last 50 epochs during the DARTS- searching for 200 epochs in S3 on CIFAR-10 dataset.

