# OpenReview forum: "DARTS-: Robustly Stepping out of Performance Collapse Without Indicators"
_ICLR.cc/2021/Conference — ICLR 2021 Poster_

### Official Review · AnonReviewer1 · 2020-10-26
**Interesting observation, but more experiments needed**

**Rating:** 6
**Confidence:** 5

**Review:**

This paper presents an interesting method to alleviate the mode collapse of DARTS (all operations degenerate to skip-connect). This is done by simply adding a skip-connect operation to complement the output of the cell function and making the coefficient of the auxiliary operation decay with time. The method is tested on a few benchmarks and settings.

Overall, I think the idea is neat and interesting, but more experiments need to be done to *support the claim of this paper*.

1. The statement that "skip-connect has two roles, one for auxiliary connection and one for candidate operation" is interesting, but not well justified. According to the authors' claim, the skip-connect operator start to dominate the cell structure after a long training procedure *because* the role of auxiliary connection becomes more and more important along with the training procedure. This is not supported by either [Zela, ICLR'20] or [Bi, 1910.11831], both of which claimed that the dominance of skip-connect comes from the incorrectly computed gradients of DARTS. *It is highly suggested to discuss this point.* Moreover, to justify the statement, an experiment is easily done: keep the auxiliary skip-connect throughout the search procedure (\beta==1), and observe if mode collapse still happens. If not, please provide the results and the guess will become more convincing.

2. The claim of "preventing mode collapse" is not well justified using experiments. For example, if the search procedure (on CIFAR10) is extended to 200 epochs or longer, is it possible to avoid mode collapse using the proposed method? It would be interesting to list a few results with varying lengths of the training procedure.

3. Some experimental results need explanation.

(1) In ImageNet experiments, the authors reported a 76.2% accuracy with 467M parameters. This really surprised me, before I checked the searched architecture and noticed that this architecture was obtained from the MobileNet space. Provided the name of DARTS-, this is VERY confusing, seemingly to guide the readers to believe that this is obtained in the DARTS space. I urge the authors to try their best to avoid such misunderstanding.

(2) (minor) In the ablation study, the comparison against P-DARTS and PC-DARTS is weird. Even without the constraint of 2 skip-connect operations, P-DARTS does not report 96.48% on CIFAR10, even lower than the random search baseline.

(3) (minor) Since a "bad" hyper-parameter of PC-DARTS (K=2) is used, I am wondering how the results are like if K=4 is used. BTW, the PC-DARTS paper reported 97.3% accuracy using K=2, but this paper reports 97.1%.


I am now neutral on this paper, slightly leaning towards rejection. Provided more complete results, I will consider changing my score.

AFTER READING THE REBUTTAL, I changed my score from 5 to 6.

---

> ### Author Response · Authors · 2020-11-15
> **DARTS- is still robust in case of non-decaying beta or longer epochs.**
>
> **Q1**: [Zela, ICLR'20] and [Bi, 1910.11831], both claimed that the dominance of skip-connect comes from the incorrectly computed gradients of DARTS.
>
> **A1**: Zela published two papers in ICLR'20, RobustDARTS (an important baseline for our paper), and NAS-Bench-1Shot1 [1]. As far as we know, there is no similar statement in these two works saying "the dominance of skip-connect comes from the incorrectly computed gradients of DARTS". Quite the contrary, RobustDARTS attributes the poor performance to falling into a sharp local minimum and inconsistency at the pruning step, based on which they propose an early-stopping strategy to avoid stepping into such a minimum and other regularization techniques to improve the generalization of pruned models.
> As per [Bi, 1910.11831], it is so far an Arxiv report. We expect the reviewer's further clarification on this particular issue.
>
> **Q2**: Keep the auxiliary skip-connect throughout the search procedure ($\beta=1$).
>
> **A2**: In the early stage of this work, it was also our first intention to keep $\beta=1$ for the auxiliary skip-connect throughout searching. We don't observe the collapse either when we searched for 3 independent times, and these models obtain an average of 97.25%$\pm$0.1 top-1 accuracy. We've appended these results in the paper (see Figure 16 in the appendix). However, there is an inconsistency issue if we don't gradually decay $\beta$ to zero: when we infer models from the supernet, the supernet still has an auxiliary skip connection while the child model doesn't. It appears that the proposed decay strategy also helps improve the searching performance (avg. 97.41$\pm$0.08%).
>
> **Q3**:  Searching on CIFAR-10 is extended to 200 epochs or longer.
>
> **A3**:  We carefully checked [Bi, arxiv 1910.11831], and we believe that it requires some hyper-parameter tuning when extending 50 to 200, since those for 50 epochs may not be feasible for longer epochs.
> Particularly, we search for 150 and 200 epochs due to strict time constraints. For these long-epoch experiments, we slightly change the schedule strategy for $\beta$, where we keep $\beta$ unchanged until we have 50 epochs left for $\beta$ decay.  Other hyper-parameters remain as they were. It turns out that DARTS- can stably handle these longer situations, and normal cells don't suffer from the aggregation of skip connections, the detailed results are listed in Table 8 (rebuttal version).
>
> It is still unclear whether longer epochs can truly boost searching performance.  Although we achieve a new state-of-the-art result in S2 where the best model has 2.50% error rate (previously 2.63%), it still has worse average performance (2.71$\pm$0.11%) in S0 than the models searched with 50 epochs (2.59$\pm$0.08%), and the best model in S3 (2.53%) is also weaker than before (2.42%).
>
> Also note that AmendDARTS  [Bi, arxiv 1910.11831] can survive in 200 epochs, but its cost is much higher than the second-order DARTS. However,  DARTS- has the same cost as the first-order of DARTS, which is more efficient.
>
> **Q4**: Avoid misunderstanding on ImageNet.
>
> **A4**:  Thanks for the suggestion and we've revised the manuscript to make it clearer. For ImageNet experiments, we mentioned in Section 4.1 that we use a popular MobileNetV2-alike search space (called S5) as in FBNet. We use this space for a fair comparison with FBNet, MnasNet, MobileNetV3, and EfficientNet. The performance collapse issue in such a search space is also reported in [2]. This result attests that DARTS- can robustly generalize in S5.
>
> **Q5**: The comparison against P-DARTS and PC-DARTS is weird.
>
> **A5**: Both are conducted with priors removed. We've made it clearer in the revision.
>
> (1) P-DARTS uses two human-designed regularization techniques: operation-level Dropout and architecture refinement (M=2). The accuracy 96.48% (3.52% error rate) is obtained by removing these regularizations. Noticeably, under a similar setting (without dropout, M=4), P-DARTS reports an error rate of 3.51%  (See the second paragraph in Section 4.4.3 of [3]).
>
> (2) PC-DARTS originally utilizes channel shuffle, which turns out to be a strong prior. To better evaluate their proposed method, we remove this prior in our experiments, e.g.,  97.1% (K=2) and 97.28% (K=4) across 3 seeds. In comparison, PC-DARTS- (K=4, combining DARTS-, no channel shuffle) obtains 97.42%.
>
> **References**
> 1. Zela et al., NAS-Bench-1Shot1: Benchmarking and Dissecting One-shot Neural Architecture Search, ICLR 2020
> 2. Chu et al., Fair darts: Eliminating unfair advantages in differentiable architecture search. ECCV 2020.
> 3. Chen et al., Progressive Differentiable Architecture Search: Bridging the Depth Gap between Search and Evaluation, ICCV 2019

---

### Official Review · AnonReviewer2 · 2020-10-27
**DARTS- review**

**Rating:** 8
**Confidence:** 5

**Review:**

This paper aims to improve the robustness of DARTS, and proposes to add an auxiliary skip connection branch to the “mixOp” in cells. The authors also analyze the effect of auxiliary branch on residual block from the view of gradient flow. Additionally, this paper refers to the theory of the work [1] and demonstrate that the auxiliary branch is able to reduce the dependence of the convergence of network weight on $\beta_{skip}$ so as to disentangle the “two-fold role” of skip connection: “as an auxiliary connection to stabilize the supernet training, and as a candidate operation to build the final network”. Moreover, extensive experiments on multiple search spaces and datasets are conducted, showing the effectiveness of this method. They also illustrate the accuracy landscape w.r.t the architecture parameters to demonstrate that the auxiliary branch is able to smoothen the landscape, making DARTS- less sensitive to the perturbation than DARTS

I have two questions:
1). I notice that the authors adopt decay strategy on the weight of auxiliary branch ($\beta$), and claim that their method is insensitive to the type of decay policy. I think the authors find an interesting phenomenon of skip connection, and I wonder how $\beta$ affects the searching procedure? And can the effect of auxiliary branch holds or how long can the effect holds after $\beta$ decays to 0. Specifically, suppose we search for 50 epochs with auxiliary branch, then we remove the auxiliary branch and search for another 50 epochs, will DARTS collapse?
2). I agree with the authors’ point that skip connection “plays two-fold roles”. However, I wonder with the auxiliary skip connection suppress the architecture parameter of skip connection ($\beta_{skip}$), so that no skip connections are chosen in the final model?

In general, I think this paper propose a simple but efficient method to alleviate the performance collapse of DARTS, the strengths and weaknesses are listed as follows:


Strengths:
1). The authors propose a simple but efficient indicator-free method to prevent skip connection from dominating the superNet. They also demonstrate the effectiveness of auxiliary branch from the view of gradient flow and the convergence of network weight.
2). Extensive experiments on multiple search spaces and datasets show the effectiveness of the method.
3). The method can combine with DARTS variants to further improve the performance.


Weaknesses:
1). I am sort of concerned that the auxiliary skip connection may suppress the weight of original skip connection. And I hope the authors will have further analysis.
2). I wonder what is the meaning of “DARTS-”since the method actually add an auxiliary skip connection stead of removing any connections or operations. I strongly suggest the authors change the name into “regDARTS” (regularized), “gradDARTS” (graduated) etc.


[1] Zhou, Pan, et al. "Theory-inspired path-regularized differential network architecture search." arXiv preprint arXiv:2006.16537 (2020).

---

> ### Author Response · Authors · 2020-11-15
> **DARTS- finds a good configuration of skip connections instead of none.**
>
> **Q1**: How does $\beta$ affect the searching procedure?
>
> **A1**: The sensitivity analysis was presented in Table 10 of A2.3 due to the page limit (submission version) and now moved to Section 4.4 in the main text (rebuttal version). It turns out that a bigger $\beta$ is more advantageous. It is as expected since the impact of the original skip connection is substantial and we need to remove its strong influence with a bigger $\beta$.
>
> **Q2**:   Can the effect of the auxiliary branch hold or how long can the effect hold after $\beta$ decays to 0 (50 epoch with the auxiliary skip and 50 without)?
>
> **A2**:  First of all, it is sufficient to stop searching with a total length of 50 epochs. Figure 4(b)  in the appendix shows that the stand-alone performance of the pruned models grows along with epochs (96.5%, 96.7%, 96.7%, 97.0%, 97.4%). When $\beta$ decays to 0, we found the best model with 97.4% top-1 accuracy.
>
> As per your question, we elongate the search to 100 epochs. For the first 50 epochs, we decay $\beta$ to zero. For the next 50 epochs, it is equivalent to applying standard DARTS to a not well-optimized supernet, which is prone to collapse in the end (as expected since it is essentially DARTS). Note the learning rate at epoch 51 is still large (near 0.012), hence it has great potential to drift away from a good solution. This is confirmed by the experiment results, where we observe an increasing number of skip connections from 2 to 6 in the second stage.
>
> **Q3**:  The auxiliary skip connection may suppress the weight of the original skip connection so that no skip connections are chosen in the final model.
>
> **A3**:   Skip connections are still chosen in the final models.  As shown in Figure 9, the best searched models contain various numbers of skip connections (e.g. the normal cell of the best model in S0 has 2), which attests that our method can indeed find a good configuration of skip connections, not just excluding all of it. It can be thought that the auxiliary skip connection mainly removes the role (1) from it, but leaves the role (2). As all operations are within the residual structure, they still have to contend to win on a fair basis.
>
> **Q4**:  Suggest to change the name into “regDARTS” (regularized), “gradDARTS”.
>
> **A4**:  Thanks for the suggestion. However, as mentioned in the title "Robustly Stepping out of Performance Collapse Without Indicators", we name our method "DARTS-" since it removes the need for human-designed indicators like Hessian eigenvalues for collapse (also indicated in footnote 1 on page 2). Although it is somewhat counter-intuitive, it would be less informative in this regard if we switch to regDARTS or its alike. We intend to call upon the community to think of indicator-free methods. As a comparison, RobustDARTS is a typical indicator-based approach that is prone to rejecting good models and is costly.

---

> > ### Comment · AnonReviewer2 · 2020-11-23
> > **respond to author feedback**
> >
> > Thanks to the detailed feedback, as my concerns are mostly addressed, I decided to stick to my initial score.

---

### Official Review · AnonReviewer3 · 2020-10-29
**interesting idea to separate the two roles of skip connection to prevent model collapse**

**Rating:** 6
**Confidence:** 3

**Review:**

The paper reveals two roles of skip connection to prevent model collapse: stabilize the supernet training, and as a candidate operation to build the final network. Intuitively, the skip connection playing the first role should only be there during the training phase. Therefore, the authors propose to add an auxiliary skip connection to play that role. This auxiliary skip connection is decayed gradually during training. The paper provides an interesting theoretical analysis that this can help prevent the gradient vanishing problem.
Compared to other approaches, the proposed one has the advantage that it does not rely on any heuristic indicator. Indeed, it is found that the existing indicator based on Hessian eigenvalue can discard good models.
In the experiments, this method - DART-, is compared with DART and several other approaches, and show that the proposed method can outperform the others, however, by small margins.

pros:
The assumption that skip connections play two different roles is very interesting and inspiring. It is indeed the case that such skip connections not only can stabilize the training, but are part of the model. The idea to separate the two roles is well motivated.
The paper provides some interesting theoretical analysis to show the potential impact of the auxiliary skip connection on the gradient vanishing problem.
The experiments are extensive, using several datasets and comparing several existing approaches. The experimental results provide some evidence that the idea works.

cons:
While the basic idea is well motivated, one could question about the specific architecture to add the auxiliary skip connection. The latter is intended to play the role of stabilizing the training. It is unclear why this specific architecture is appropriate for it. The paper does not provide a strong explanation for it.
While DART- outperforms many of the baselines, it is only marginally better than some recent methods (namely P-DART). For example, in Table 2, DART- is equivalent to P-DART on CIFAR-10.  The experimental evident is moderately strong to show that DART- is better than the existing methods.

Recommendation to the authors: It would be useful to justify the choice of the architecture of DART-, namely to explain why the auxiliary skip connection added can truly play the role of stabilizer. Would it be possible to use a different architecture as well?

---

> ### Author Response · Authors · 2020-11-15
> **DARTS- is more robust in terms of searching performance.**
>
> **Q1**: DARTS- is only marginally better than some recent methods (namely P-DARTS).
>
> **A1**: There is a legacy misunderstanding since DARTS originally reported its average accuracy of the best model when trained several times, so are many of its variants. This only shows the training stability of a single model. However, it is recommended to evaluate NAS using the averaged performance while searching independently for several times [1,2,3]. We've also made this point clear in the footnote of Table 2.
>
> P-DARTS reported the best accuracy (97.5%) of a single model on CIFAR-10 in S0. To meet the above-recommended evaluation standard, we run P-DARTS's searching code several times to obtain an average of 97.19$\pm$0.14%. To get a better model as good as reported, one might need to search many times, adding additional costs for both searching and training. In contrast, DARTS- obtains 97.41$\pm$0.08%, which is more stable than P-DARTS.
>
> Besides, P-DARTS relies on two human-designed regularization techniques: limiting the number of skip connections and applying operation-level dropout. When these priors are removed, P-DARTS's performance falls to 96.48$\pm$0.55 (see Table 4)%.  It suggests that these two priors matter more than the progressive process. Furthermore, to evaluate the effectiveness of our method, we also combine our indicator-free approach DARTS- with prior-removed P-DARTS (i.e. P-DARTS-), it can still obtain 97.28$\pm$ 0.04%. Among these options, our standalone method is the most robust.
>
> Lastly, the human-designed regularizations in P-DARTS may require extensive hyper-parameter tuning, it is more so given different benchmarks.  For instance, they wrote, "The initial Dropout probability on skip-connect for the reported results is set to be 0.0, 0.4, 0.7 on CIFAR-10 for stages 1, 2 and 3, respectively, and 0.1, 0.2, 0.3 for CIFAR-100."  In contrast, our method is quite robust and doesn't require tedious tweaking. DARTS- mostly obtains better results compared with previous art across 12 benchmarks (3 datasets and 7 search spaces) with the same  $\beta=1.0$.
>
> **Q2**: It would be useful to justify the choice of the architecture of DARTS-, would it be possible to use a different architecture as well other than skip connection?
>
> **A2**:  Thanks for the recommendation. This ablation study was included in appendix A.2.3 regarding page limitation and now moved to Section 4.4 in the main text (rebuttal revised version).
>
> Skip connection is a simple and proper choice as we let it assume the role to stabilize the supernet training, however one can think of other options. Specifically, we use a 1x1 convolution with weights initialized as an identity matrix, but later be learnable during optimization. Albeit being effective (97.25$\pm$0.09%), it turns out to be worse than skip connection (97.41$\pm$0.08%). We consider it is due to that the learning process breaks its similarity to a residual structure, the 1x1 projection can no longer take up the role as skip connection does. Therefore, we use the skip connection as our default choice, which is similar to He's choice in ResNet [4].
>
> **References**
> 1. Yang et al., NAS evaluation is frustratingly hard, ICLR 2020
> 2. Zela et al., Understanding and robustifying differentiable architecture search, ICLR 2020
> 3. Chen et al.,  Stabilizing differentiable architecture search via perturbation based regularization, ICML 2020
> 4. He et al., Deep Residual Learning for Image Recognition, CVPR 2016

---

### Official Review · AnonReviewer4 · 2020-10-29
**Addressing an important problem in neural architecture search**

**Rating:** 6
**Confidence:** 2

**Review:**

The paper focuses on improving the robustness of differentiable architecture search models. I am not an expert in the topic, so please see my review as an outsider's perspective. Although the problem is clearly formulated and motivated, I don't really understand the actual contribution of the paper. There is no intuitive explanation of the approach and why it should work. I hope the other reviewers are expert in the topic and can comment on this aspect, but as an outsider, I think the paper is poorly written (the English is clear, it's just the way they explain the solution and contribution) and the contributions are not clear.

I tried to look at the prior work (mostly those mentioned in the paper and frequently cited) and based on this quick research, it seems that the approach is novel. However, again, I do believe that at least an expert view on this aspect is necessary. I am not sure if I know all the prior work.

The experimental results are encouraging. They show robustness across datasets. I think this is a strong paper from an empirical perspective.

Since I do not fully understand the contributions and the authors didn't convince me on why this is a novel approach, I'd vote for marginally above acceptance threshold (mainly because of strong evaluation).

---

> ### Author Response · Authors · 2020-11-15
> **An intuitive explanation of our contribution.**
>
> **Q1**: The English is clear, but the contributions are not clear.
>
> **A1**: Thank you for your valuable time in reviewing our paper. We wish that we have made our contributions clearer also for the experts not in this scope of research.
>
> In a nutshell, our contribution is to provide a novel indicator-free framework (as our title indicates) to solve a well-known performance collapse issue in differentiable architecture search. Our solution is more robust and nearly adds no extra cost compared with previous works. For instance, [1,2] use time-consuming indicators like Hessian eigenvalue (calculated at every epoch to know whether the collapse is about to happen or not), which takes almost the same amount of time as training the supernet for one epoch. Apart from the cost, it is also hard to design a perfect indicator of the collapse. For instance, a higher Hessian eigenvalue not necessarily leads to poor models. Please see Fig.4 (b) in the appendix for the details. That being said, an improper indicator can avoid bad models but might reject good models. In this paper, we tried to reason what has caused the collapse in the first place and then regularize the training with a simple but effective technique.
>
> Specifically, we are motivated to address the collapse based on our observation of the two-fold roles that skip connections play in the DARTS supernet:
> -  as a connection to stabilize the supernet training,
> - as a candidate operation to build the final network.
>
> We propose to use an auxiliary skip connection to carry out the former role and leave the original skip connection to take up the latter. Besides, we gradually decay the impact of the auxiliary connection during the search. At the pruning step, the supernet can be seen as the one without this auxiliary connection, which is more consistent with the pruned model. Extensive experiments show that our method successfully resolves the collapse of DARTS across a large body of important NAS benchmarks. Due to the uttermost simplicity and generality, our approach can also be applied to a variety of DARTS variants by replacing the human-designed indicators to boost the performance (e.g. P-DARTS, PC-DARTS as investigated in the paper.)
>
> **References**
> 1. Zela et al., Understanding and robustifying differentiable architecture search, ICLR 2020
> 2. Chen et al.,  Stabilizing differentiable architecture search via perturbation based regularization, ICML 2020

---

### Author Response · Authors · 2020-11-17
**A summary of updates in PDF.**

Hi all,

We thank everyone for the dedicated reviews. Here is a summary of updates in the paper,

1. Settings of P-DARTS and PC-DARTS experiments are revised in Section 4.3 (regarding doubts from Reviewer 1).
2. **Sensitivity analysis of $\beta$** moved from appendix to Section 4.4 (proposed by Reviewer 2).
3. **The choice of auxiliary branch** moved from appendix to Section 4.4 (noted by Reviewer 3).
4. **Performance with longer epochs** newly written in Section 4.4 (suggested by Reviewer 1).
5. Genotypes of longer epochs (Figure.16-21) are newly added to the Appendix.

---

### Decision · Program_Chairs · 2021-01-07
**Final Decision**

**Decision:**

Accept (Poster)

**Comment:**

The paper proposed an interesting method to improve the robustness of DARTS and hence to alleviate the mode collapse. The idea consists of adding an auxiliary skip connection branch that complements the output of the cell function together with a depth analysis about the effect of the auxiliary branch. The proposed approach is validated on a few benchmarks showing the effectiveness of the proposed approach. All reviewers agreed that the idea is simple, efficient and interesting. author response satisfactorily addressed most of the points raised by the reviewers, and most of them increased their original score accepting the paper. Therefore, I recommend acceptance.